# Combinatorial Effect of Magnetic Field and Radiotherapy in PDAC Organoids: A Pilot Study

**DOI:** 10.3390/biomedicines8120609

**Published:** 2020-12-14

**Authors:** Luca Nicosia, Filippo Alongi, Silvia Andreani, Ruggero Ruggieri, Borislav Rusev, Beatrice Mantoan, Rita Teresa Lawlor, Antonio Pea, Aldo Scarpa, Linda Agolli, Vincenzo Corbo, Sabrina D’Agosto

**Affiliations:** 1Advanced Radiation Oncology Department, IRCCS Sacro Cuore Don Calabria Hospital, Cancer Care Center, 37024 Negrar, Italy; luca.nicosia@sacrocuore.it (L.N.); filippo.alongi@sacrocuore.it (F.A.); ruggero.ruggieri@sacrocuore.it (R.R.); beatrice.mantoan@sacrocuore.it (B.M.); 2Medicine Faculty, University of Brescia, 25121 Brescia, Italy; 3ARC-Net Research Centre, University and Hospital Trust of Verona, 37134 Verona, Italy; silvia.andreani@univr.it (S.A.); borislavchavdarov.rusev@univr.it (B.R.); ritateresa.lawlor@univr.it (R.T.L.); aldo.scarpa@univr.it (A.S.); 4General and Pancreatic Surgery Department, Pancreas Institute, University and Hospital Trust of Verona, 37134 Verona, Italy; antonio.pea@univr.it; 5Department of Diagnostics and Public Health, University and Hospital Trust of Verona, 37134 Verona, Italy; sabrinaluigia.dagosto@univr.it; 6Department of Radiation Oncology, Faculty of Medicine and University Hospital Carl Gustav Carus, Technische Universität, Fetscherstraße 74, 01307 Dresden, Germany; linda.agolli@gmail.com; 7OncoRay—National Center for Radiation Research in Oncology, Faculty of Medicine and University Hospital Carl Gustav Carus, Technische Universität Dresden, Fetscherstraße 74, 01307 Dresden, Germany

**Keywords:** patient-derived models, 3D organoids, MR-guided radiotherapy, MR-Linac, radiotherapy, magnetic field

## Abstract

Pancreatic ductal adenocarcinoma (PDAC) is highly refractory to systemic treatment, including radiotherapy (RT) either as alone or in combination with chemotherapy. Magnetic resonance (MR)-guided RT is a novel treatment technique which conjugates the high MR imaging contrast resolution to the possibility of re-adapting treatment plan to daily anatomical variations. Magnetic field (MF) might exert a biological effect that could be exploited to enhance radiation effect. The aim of the present study was to lay the preclinical basis of the MF effect by exploring how it modifies the response to radiation in organoid cultures established from PDAC. The short-term effect of radiation, alone or in combination with MF, was evaluated in patient-derived organoids (PDOs) and monolayer cell cultures. Cell viability, apoptotic cell death, and organoid size following exposure to the treatment were evaluated. PDOs demonstrated limited sensitivity at clinically relevant doses of radiation. The combination of radiation and MF demonstrated superior efficacy than monotherapy in almost all the PDOs tested. PDOs treated with combination of radiation and MF were significantly smaller in size and some showed increased cell death as compared to the monotherapy with radiation. Long-time exposure to 1.5T MF can increase the therapeutic efficacy of radiation in PDAC organoids.

## 1. Introduction

Pancreatic ductal adenocarcinoma (PDAC) is a lethal disease due to late presentation and poor response to available treatment modalities, including radiotherapy alone or in combination with chemotherapy [1]. As it stands, there is an urgent need to change the current standard of care through the definition of novel therapeutic options, which are ideally tailored to individual patients or patient’s subtypes, as well as through the application of modern treatment techniques. While molecular subtypes were identified through genome-wide analyses [2,3,4,5], personalized treatment is far from being implemented for PDAC. At least in part, this is due to inappropriate preclinical modeling of the disease, which has made difficult extrapolating results from model systems to humans in the past. Three-dimensional in vitro culture of human cancer cells, termed organoid, has the potential to overcome this bottleneck. Organoids have been established from a variety of normal and neoplastic human tissues [6,7,8,9,10,11,12,13,14], including pancreas [15,16], and proved to be a reliable preclinical tool to assess therapeutic responses [17,18]. Recently, Patient-derived organoids (PDOs) have also been used to test efficacy of radiation. In particular, Yao et al. demonstrated that responses to radiation observed in PDOs from locally advanced rectal cancer mirror those observed in patients, suggesting the possibility of using PDOs to guide therapeutic management of rectal cancer patients [19]. PDAC organoids can be reliably established (success rate > 70% [17]) from resected specimens as well as from fine-needle aspiration biopsies within a timeframe that is compatible with precision medicine approaches.

Radiotherapy (RT) encompasses a broad spectrum of treatment modalities and it is a rapidly evolving field with dramatic advances observed in delivery method [20]. Image-guided radiotherapy has, indeed, significantly improved the compliance of the radiation dose deposition in the tumor area while increasingly sparing healthy tissues [21,22,23]. Magnetic resonance (MR) imaging is routinely used for diagnostic purpose using 1.5–3 T static magnetic field (MF). Magnetic field might exert a physical effect, as well as temperature and pressure do, nevertheless the evidence is ambiguous. Previous experiences reported that several parameters might influence MF effect on living tissues, such as field intensity, exposure duration, and direction [24,25,26,27,28,29,30]. Moreover, biological factors such as cell type might influence MF effect [31,32]. However, these experiences showed conflicting results.

Recently, 1.5 T MR-linac was introduced in the clinical practice. This is a new type of RT device, approved for clinical use that conjugates the high contrast resolution of MR imaging to the possibility to re-adapt the treatment plan to the daily anatomical variation, with the aim to increase the accuracy of the treatment delivery [33]. Due to its structural characteristics, patients are treated into the MR gantry and exposed daily to MF before and during fractionated RT schedules. Treatment duration on a 1.5 T MR-linac may vary between 40 and 60 min depending on the specific workflow required by MR-linac, which consists of MR imaging acquisition, region of interest contouring, replanning, pretreatment imaging verification, and treatment delivery [34].

In clinical practice, patients and operators may be normally exposed to MF for short time and therefore its potential biological action is considered negligible. Nevertheless, in the case of prolonged and repeated MF exposure, as during the treatment on a MR-linac, it might exert a detectable biological effect [22].

This ability might be exploited to enhance the cytocidal effect of radiation. Therefore, the identification of such biological effect might have interesting therapeutic implications, since MR would be used not only for imaging purposes but also as a novel potential treatment option. Very limited experiences reported on the therapeutic association between RT and MF, with the majority of them showing minimal or no detectable effect [22,23]. These studies are limited by the use of weak or induced magnetic fields, or by a limited (or not reported) exposure time [23,35]. Therefore, whether this effect exists and can be controlled to influence tumor cell growth remains unclear. The biological effect of the magnetic field is revealed by an increased apoptosis and seems dependent on field intensity [25], but very limited evidence gives additional clue. In this study, we assessed the preclinical basis of the interaction between static magnetic field and radiation in organoid cultures established from treatment-naïve PDAC.

## 2. Materials and Methods

### 2.1. Patient Samples

Pancreatic cancer tissues were obtained from patients undergoing surgical resection for curative intent at the University and Hospital Trust of Verona. All human experiments were approved by the local Ethical Committee of the University of Verona, Italy (Prot. CESC 50401, 28 October 2015). Written informed consent from the donors for research use of tissue in this study was obtained prior to acquisition of the specimen. Samples were confirmed to be tumor based on pathological assessment. All experiments were conducted in accordance with relevant guidelines and regulations.

### 2.2. Generation of PDAC Cultures

A total of nine PDAC cultures were used in this study (Table 1). PDAC organoid cultures (*n* = 6) were established as previously described [15]. Briefly, tumor specimens were minced and digested with Collagenase II (5 mg/mL, Gibco) and Dispase I (1.25 mg/mL, Gibco) in human complete medium [15] at 37°C for a maximum of 2 h. The resulting material was further digested with TrypLE (Gibco) for 10 min at 37°C, embedded in growth factor reduced Matrigel (Corning) and cultured in human complete medium (described in [15]). Tissue digestions from three additional PDAC specimens were directly seeded on tissue-culture vessels for initiation of monolayer cell cultures using the following medium: Advanced DMEM/F12 medium supplemented with HEPES (1X, Gibco), Glutamax (1X, Gibco), Primocin (1 mg/mL, Invivogen), mouse Epidermal Growth Factor (50 ng/mL, Gibco), Dexamethasone (3 nM, Sigma), and 5% Fetal Bovine Serum (Gibco). Both monolayer cell cultures and organoids were routinely tested for the presence of mycoplasma using MycoAlert Detection Kit from Lonza in accordance with the manufacturer’s instructions.

### 2.3. Histology and Immunostaining

Tissues and organoids were formalin-fixed and paraffin-embedded using standard procedures. Organoids were harvested and incubated with a Dispase solution (2 mg/mL Dispase I) for 1 h to digest the matrix, washed in DPBS (Gibco), fixed in 10% formalin for 30 min, followed by incubation in 70% ethanol for 10 min. Then, organoids were centrifuged at 200× g for 5 min at 4 °C and resuspended in HistoGel (Thermo Fisher Scientific, Milan, Italy) before tissue processing. Haematoxylin-eosin staining was performed on 3µm paraffin sections. For immunostaining, the following primary antibodies were used: KRT7 (Clone RN7, Leica, Milan, Italy) 1:100; KRT20 (Clone PW31, Leica) 1:2000; MUC1 (Clone Ma695, Leica) 1:100; MUC2 (Clone Ccp58, Leica) 1:100; MUC5AC (Clone CLH2, DAKO) 1:50.

### 2.4. Mutational Analysis by Next-Generation Targeted Sequencing

DNA was extracted from PDAC tumor tissues and organoid cultures using the DNAeasy blood and tissue kit (QIAGEN) and quantified with the Nanodrop Spectrophotometer (Thermo Fisher Scientific). A multigene panel was used to investigate mutational status of 20 genes (KRAS, TP53, SMAD4, ATM, APC, BRAF, CDH1, CDKN2A, CTNNB1, EGFR, ERBB2, ERBB4, FBXW7, FGFR3, FLT3, GNAS, HRAS, KDR, NRAS, and PIK3CA). Twenty nanograms of DNA were used for multiplex PCR amplification. The quality of the obtained libraries was evaluated by the Agilent 2100 Bioanalyzer on-chip electrophoresis (Agilent Technologies, Santa Clara, CA, USA). Emulsion PCR to construct the libraries of clonal sequences was performed with the Ion OneTouch™ OT2 System (Thermo Fisher Scientific). Sequencing was run on the Ion Proton (PI, Thermo Fisher Scientific, Milan, Italy) loaded with Ion PI Chip v2. Data analysis, including alignment to the hg19 human reference genome and variant calling, was done using the Torrent Suite Software v.5.0 (Thermo Fisher Scientific, Milan, Italy). Filtered variants were annotated using a custom pipeline based on vcflib (https://github.com/ekg/vcflib), SnpSift, the Variant Effect Predictor (VEP) software and NCBI RefSeq database. Additionally, alignments were visually verified with the Integrative Genomics Viewer (IGV) v2.3 to further confirm the presence of mutations identified by targeted sequencing.

### 2.5. Therapeutic Experiments

For cell viability assay, organoid cultures were first released from Matrigel by incubation with a solution of Dispase I at 37 °C for 20 min, and then subjected to enzymatic digestion with TripLE supplemented with Dispase I and 0.1mg/mL DNAse I (Sigma-Aldrich, Milan, Italy) for 20 min. Single cells were counted and diluted to obtain 10 cells/µL in a mixture of human complete medium, Rho Kinase inhibitor (10.5 µM, Sigma-Aldrich, Milan, Italy) and Matrigel (final concentration 10%). 100 µL of this mixture was plated in individual wells of 96-well plate (Nunc, Thermo Fisher Scientific, Milan, Itlay). For cell imaging, organoids were harvested using ice-cold Cell Recovery Solution (Corning) and incubated for 1 h on ice to dislodge the matrix. Organoids were then triturated in ice-cold medium through a fire-polished glass pipette (Corning) and resuspended in Matrigel. Organoids density was adjusted before seeding in 50 µL Matrigel to obtain 105 single cells in each well of a 24 well plate (Greiner). Finally, Matrigel containing cells was overlaid with 500 µL of culture medium. Cultures were exposed to radiation (X-Rays and MF alone or in combination) 40 h after plating to allow organoids reformation. Two, 4, 6, 8, 10 and 12 Gy were the doses of irradiation used with five replicates per dosage. For cell viability, adherent primary cells were dissociated to single cells following incubation with trypsin (1x, Gibco) at 37 °C for 2 min, counted and diluted to obtain 20 cells/µL in 100 µL of culture medium. 100 µL of cells-containing medium was plated into individual wells of a 96-well plate (Thermo Fisher Scientific). A dose of 6 Gy and a minimum of three replicates per cell cultures were used. Seventy-two hours after treatment, cell viability of organoids and 2D cell cultures was measured using the CellTiter-Glo^®^ Cell viability assay (Promega, G9683) according to manufacturer’s instruction. Organoids and 2D cultures were also photographed using EVOS Cell Imaging System (Thermo Fisher Scientific) at the same time point (72 h post irradiation). For evaluation of apoptotic cell death, organoid cultures were pre-incubated (24 h before assessment, 48 h post treatment) with a fluorogenic substrate of Caspase-3/7 (CellEventTM Caspase-3/7 Green Detection Reagent, Thermo Fisher Scientific, Milan, Italy). Monolayer cell cultures were incubated with CellTrace Calcein Green AM (Thermo Fisher Scientific) 30 min. For evaluation of organoid size change, images of live organoids following 6 Gy exposure were processed with ImageJ software. Exposure of culture plates (either 2D or 3D cell cultures) to the treatments was performed at room temperature in ambient atmosphere (no temperature variation was observed before and after treatment). Accordingly, untreated controls (i.e., cultures not exposed to MR or RT) are matched cultures kept in the same condition for the same duration of the treatment.

### 2.6. Phantom Design

We designed a phantom for treatment delivery constituted by 10 square slabs of solid water (RW3, PTWTM) with 30 × 30 × 1 cm (W-L-H) size each, with equivalent-to-water attenuation properties in the used MV (Megavolt) photon energy range. Two cell culture plates (W-L-H: 10 × 4 × 1 cm each) were placed above the center of the tenth overlaid slab, with a water-equivalent bolus placed all around such plates to assure lateral electronic equilibrium. Three further solid water slabs (RW3) were posed above the cell cultures to assure a cells’ depth larger than the build-up depth (Figure 1).

### 2.7. Phantom Simulation and Treatment Plan

A CT (computed tomography) simulation scan with a slice thickness of 3 mm, for electron density assignment, followed by a fast 3D T2-weighted FFE (fast field echo) simulation MRi (MRs) with 1 mm slice thickness, the same scan used daily for treatment by the 1.5 T MR-Linac, were acquired for dose calculation purposes. The 1.5 T MR-Linac (UnityTM, Elekta Inc., Sweden) is a hybrid machine consisting of a linac producing a 7MV flattening filter free (FFF) photon beam, together with a 1.5 T MR unit. To assure homogeneous cell dose delivery, a plan with two opposed AP-PA fields was prepared by taking as target (PTV) a 1.5 cm expansion of the structure constituted by the summation of the two cell plates. For such PTV, at least 95% of the prescribed dose (Dp) was assured to at least 95% of the PTV, while less than 2% of PTV (D2%) could receive a higher than 107% Dp dose, consistently with International Commission for Radiological Units (ICRU) recommendations. The in vitro tested Dp for organoid cultures was 6 Gy in single fraction. Before each treatment, a fast 3D T2-weighted FFE MRi scan (pre-MRi) was acquired and used to adapt the plan of the day to compensate for positioning errors. Additionally, 2D cell cultures were treated on a conventional linac. Before each treatment, a cone-beam CT was acquired, and positioning error thus corrected.

### 2.8. Dose Delivery

Organoid cultures were treated by 6 Gy. Four cell plates were prepared as follows: control (plate 1), MR alone (named MF) (plate 2), radiotherapy alone (named X-rays) (plate 3), MR plus radiotherapy (named combo) (plate 4). Treatment workflow is summarized hereafter: plates 2 and 4 were exposed to SMF for 60 min. Afterwards, plate 2 was replaced by plate 3 and radiation treatment was delivered. Two-D cell culture were treated with the following radiation doses: 2, 4, 6, 8, 10 and 12 Gy.

### 2.9. Statistical Analysis

Quantitative data are expressed as the mean ± SD. Comparisons were conducted using student *t* test. A *p* < 0.05 indicated a statistically significant difference. GraphPad Prism 8 software was used for graphing.

## 3. Results

### 3.1. Establishment and Characterization of PDAC-PDOs

Six PDAC organoids were established from treatment-naïve patients and displayed heterogenous morphology in culture (Table 1 and Figure 2A). In particular, PDAC1 grew as cystic organoids while PDAC2 and PDAC3 formed both cystic and solid organoids (Figure 2A). Haematoxylin and eosin staining of paraffin-embedded organoids showed that tumor-derived organoids present with different grade of differentiation ranging from moderately differentiated (PDAC1) to well-differentiated carcinomas (PDAC4) (Figure 2B). To further investigate the histopathological feature of tumor organoids and how this relates to parental tissues, we performed immunohistochemistry for pancreatic markers KRT7, MUC1 and MUC5AC, and the intestinal markers KRT20 and MUC2. As expected, we observed high immunoreactivity for KRT7 in the cytoplasm of PDAC1 organoid culture (Figure 2C), and no immunopositivity for KRT20 and MUC2. Staining for mucins demonstrated immunopositivity for MUC1 and MUC5AC in the lumen of the cells composing the organoids (Figure 2C). As previously reported [15,17], PDOs preserved alterations in driver genes of the corresponding tissues, which is a fundamental requirement for a proper preclinical model (Figure 2D). All cultures harbored KRAS alterations, 4 out of 6 carried TP53 mutations, and only two carried SMAD4 inactivating mutations.

### 3.2. Responses of PDOs to Irradiation

To determine the response of PDAC-PDOs to irradiation in vitro, we performed a short-term viability assay measuring ATP-cellular content 72 h after exposure of organoids to increasing doses of X-Rays (2, 4, 8, 10, and 12 Gy). The 6 PDAC-PDOs tested displayed heterogeneous responses to irradiation (Figure 3A). In particular, radiation showed no effect on PDAC2 and PDAC4 even at high doses, and only one culture (PDAC3) exhibited a significant reduction in viability when exposed to 10 and 12 Gy. Overall, PDOs showed limited radiosensitivity at clinically relevant doses of radiation. Next, we sought to assess whether concurrent exposure to X-Rays and MF had higher efficacy than monotherapy in reducing PDOs viability. To this aim, we selected and tested four PDOs showing different range of sensitivity to X-Rays. While MF alone had minimal effect on viability of 3 out of the 4 PDOs tested, the combination of 6 Gy and MF further reduced cell viability in all cultures as compared to monotherapy with X-Rays (Figure 3B). Reduction in ATP-cellular content might be due to reduced cell proliferation under specific condition rather than cell killing. Therefore, we measured induction of apoptotic cell death following exposure to radiation alone or in combination with MF by incubating organoid cultures with a fluorogenic substrate for the activated executioners Caspases 3 and 7 (Figure 3C). In keeping with ATP-based measures, combination of X-Rays and MF significantly increased cell death of 59% and 33% compared to control while reducing organoids size in PDAC1- and PDAC2-PDO, respectively (Figure 3D,E). No induction of apoptotic cell death could be observed in PDAC4 culture following treatment, yet organoids reduced in size (Figure 3D,E). Overall, PDAC-PDOs displayed heterogeneous responses both to the radiation and to the MF, thus suggesting that not all patients are likely to respond to the treatment. Sensitivity could not be cross-referenced to peculiar morphological or genetic characteristics of the PDOs. PDAC4 showed the least sensitivity and, interestingly, it was derived from a patient who has been treated with adjuvant radiotherapy, but treatment was soon suspended as the patient rapidly progressed.

### 3.3. Viability of 2D PDAC after Exposure to Irradiation

Next, we sought to investigate whether irradiation has a direct impact on cell viability of PDAC cells or rather induces humoral responses that drive cell death. To this aim, we employed patients’ derived 2D primary cultures (Table 1), which were exposed to X-Ray as monotherapy. Twenty-four hours after seeding, cells were treated with 6 Gy and cell viability evaluated 72 h later as described before. To test whether radiation-induced secretion of molecules that might affect cell viability, the medium was either replaced immediately after treatment or maintained in culture (Figure 4A). Among the three cell lines tested, PDAC7 showed the highest sensitivity to the treatment with no difference observed when media was maintained in culture or replaced. On the other hand, only a minimal decrease in viability was observed for PDAC8 and PDAC9 when culture media was replaced immediately after treatment, suggesting that irradiation determines the release of molecules that might affect cell viability (Figure 4A). To further test this hypothesis, we exposed untreated cell lines to conditioned media from cell lines (either matched or unmatched) previously exposed to radiation and observed a reduction in cell viability as assessed by both ATP-based measures and imaging (Figure 4B–D).

## 4. Discussion

The implementation of new technologies in medicine questions on how new clinical applications can be pursued to increase the available therapeutic armamentarium. In last years, MR-linacs are gaining increasing interest due to the particular characteristics of treatment plan adaptation and high image quality that might increase radiation treatment accuracy, effectiveness, and tolerability [33,36]. MR-linac is characterized by the structural presence of a high-field magnet to which patients are daily exposed for a long time. Therefore, a question has arisen on whether MF might have a biological effect, how this would be eventually exploited to increase radiation effectiveness, and how would play a role in normal tissue response to treatment. Several preclinical studies have explored the effect of MF on cellular processes and phenotypes. Cells can sense and respond to MF, and the heterogeneous responses observed in cellular systems depend on several factors, which includes the characteristics of both MF (e.g., time, intensity, and type) and cells (e.g., age, cell type, and cell status) [22]. Collectively, static MF has been shown to influence multiple cellular features, including orientation of cells and biomolecules [37,38,39], cell proliferation [40,41,42], cell attachment/adhesion [43,44], cell morphology [32,33,34,35,36,37,38,39,40,41,42,43], cell cycle [41,42,43,44,45] and migration [46], and the production of reactive oxygen species [47,48].

As an example, Zhang et al. showed that static magnetic fields inhibit EGFR kinase activity with consequent inhibition of cell proliferation [49]. Furthermore, MF exposure can act on the mitotic spindle by affecting the orientation and the morphology in a field intensity- and time-dependent modality [50], and by increasing the rate of abnormal spindles [51]. Tian et al. [29] demonstrated that not only MF exposure but also the direction of the field (upward versus downward) can have different effect on cancer cell growth. However, only a few preclinical studies explored a potential association between MF and radiations, far from any realistic application in oncology. Li et al. [52] described an increased apoptosis in different cell lines of human hepatoma after 30 min long exposure to 0.2 T static MF generated by a natural magnetic stone. Data show that the exposure to magnetic field induced up-regulation of Caspase 9 and down-regulation of Bcl-2 expression, resulting in higher level of apoptosis. Moreover, Li and colleagues also demonstrated heterogeneous responses of different 2D cell lines exposed to the same MF intensity, which suggests the need for the identification of predictive biomarkers.

Here, we used the 1.5 T MF integrated in the MR-linac, with an exposed time 60-min long, demonstrating that MF adds therapeutic effects when administered before X-rays in all the 4 PDOs tested. In particular, PDOs displayed heterogeneous responses to X-rays exposure and were generally resistant to clinically relevant doses. The sole exposure to the magnetic field had limited effect on proliferation of PDOs, yet we could observe inter-PDOs differences. However, in all PDOs tested the priming with MF resulted in higher efficacy of X-rays in reducing cell proliferation. Given the high inter-cultures variability, the relatively small sample size is a limitation of our study. Indeed, a larger array of molecularly characterized PDOs is required to conclusively define the relevance of the combinatorial treatment as well as to potentially identify molecular signatures predictive of therapeutic responses.

The reduction in cell proliferation upon combination of MF and X-rays was associated with increase apoptotic cell death in a subset of cultures and to the reduced size of individual organoid in all cultures tested, a latter being a parameter previously used to assess efficacy of the treatment [19]. Interestingly, we also noted using monolayer cell cultures that the reduction in cell viability is at least in part due to the humoral responses from cancer cells highlighting the possibility that radiation is working through mechanisms other than direct induction of DNA damage.

The use of PDOs has already been demonstrated useful for the implementation of precision oncology by enabling the alignment of preclinical and clinical platforms to guide drug intervention [17,18,53,54]. The ability of PDAC-PDOs to predict responses to radiotherapy remains unknown, while PDOs from locally advanced rectal cancer were able to predict responses to chemoradiation in patients (90% of matching responses between patients and models) [19]. Two recent studies explored the association between MF and radiations in tumor cell lines. In the study from Yudshitara et al. [23] human TK6 lymphoblastoid cell lines were irradiated with 1 to 4 Gy after MF exposure. The results showed no increased apoptosis, concluding that no effect from MF would be expected. The authors used a MF induced by electric current through a pair of magnetic coils generating up to 1.5 T strength. Unfortunately, they did not report the cell culture MF exposure time. Similarly, Tambasco et al. [35] tested whether the presence of a 1.5 T MF increased the radiation-induced DNA damage rate on pBR322 plasmid DNA (measured as the increase in single- and double-strand breaks). The study failed to demonstrate any effect. Interestingly, radiation was administered without any magnetic priming (“magnetization”) before the treatment with X-Ray, probably suggesting that the biological mechanism underlying MF effect did not relate to a direct DNA damage. Here, we reproduced the typical treatment workflow on a 1.5 T MR-linac, with a MF exposure time 60-min long showing that the combination of MF and X-rays is associated with higher preclinical responses compared to the radiation alone in subset of PDOs. The reduced cell viability of PDOs subjected to combined MF and X-rays was associated with reduced organoid size and increased apoptotic fluxes in some cultures. No specific mechanisms for the MF effect on cell cultures has been actually identified, leaving this fundamental question on the trail. Future studies should confirm the present evidence and investigate the mechanisms underlying this biological effect. To the best of our knowledge, the present study is the first to demonstrate the preclinical basis of a potential clinical application of MF in radiation oncology, and to postulate a synergistic effect when combined with radiations. A future study has been planned to investigate the molecular basis of such effect. Whether confirmed, magnetic-enhanced radiotherapy might be explored in a clinical context aiming to improve the therapeutic ratio of radiations.

## 5. Conclusions

PDOs from treatment-naïve PDAC are relatively resistant to single irradiation using clinically relevant doses. Despite the small sample size, we observed interpatient variability in the response to radiation and no clear association between response and genetic features. While minimally affecting cell viability as monotherapy, a high-intensity (1.5 T) static magnetic field significantly increases radiation effectiveness in PDOs. The understanding of the biological mechanisms underlying this effect is currently ongoing at our institution. Whether confirmed, the clinical application of magnetic-enhanced radiotherapy will be explored.

## Figures and Tables

**Figure 1 biomedicines-08-00609-f001:**
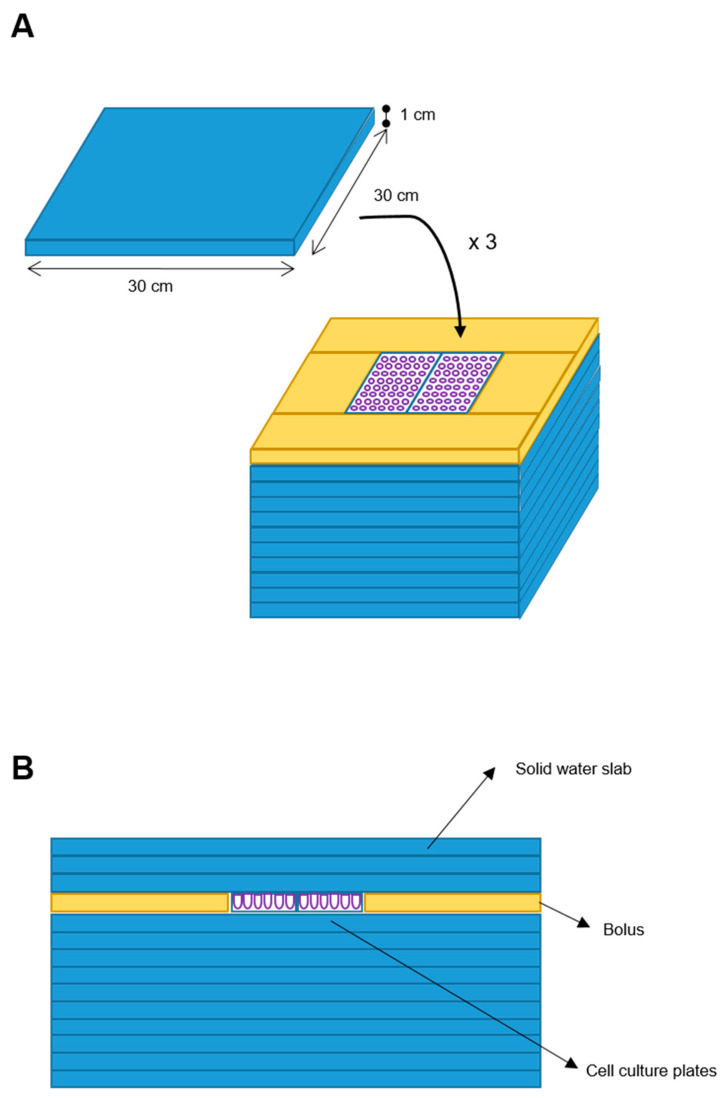
Blueprint of the phantom. Ten solid water slabs (30 × 30 × 1 cm) (blue element) were the phantom base. Two cell plates were placed at the center of the upper slab and a bolus (yellow element) was placed around. Additional three solid water slabs were added on top (**A**). Frontal sectional view of the phantom (**B**).

**Figure 2 biomedicines-08-00609-f002:**
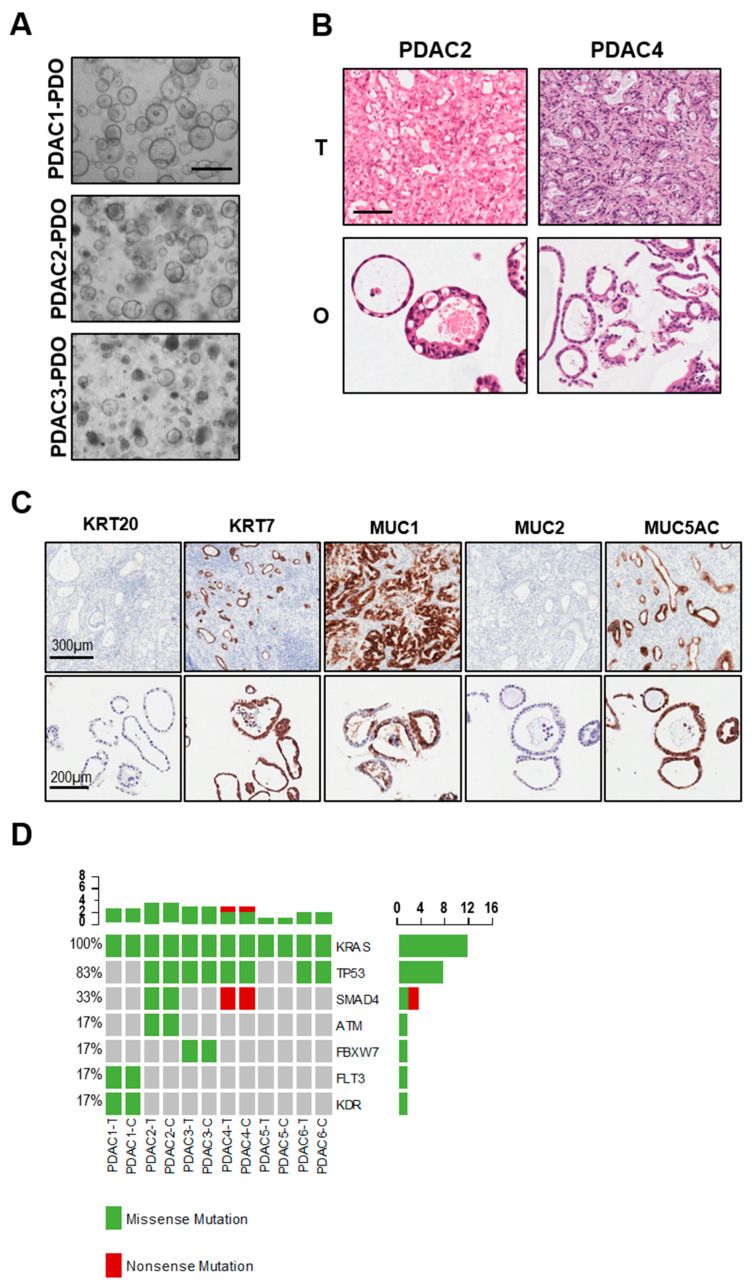
Histopathological and molecular characterization of PDAC-PDOs. (**A**) Bright-field images of established PDAC-PDOs. Scale bar, 500 μm. (**B**) Haematoxylin and eosin stained sections of PDAC organoids and their parental tumor tissues. Scale bar, 200 μm. (**C**) Histopathological characteristics of PDAC1 organoids and their corresponding tumor tissues. Immunohistochemistry for KRT20 (negative), KRT7 (positive), MUC1 (positive), MUC2 (negative), MUC5AC (positive) of paraffin-embedded organoid culture and matched parental tumor tissue. (**D**) Non-synonymous somatic mutations found in PDAC organoids and their parental tumor tissues through high-coverage targeted sequencing.

**Figure 3 biomedicines-08-00609-f003:**
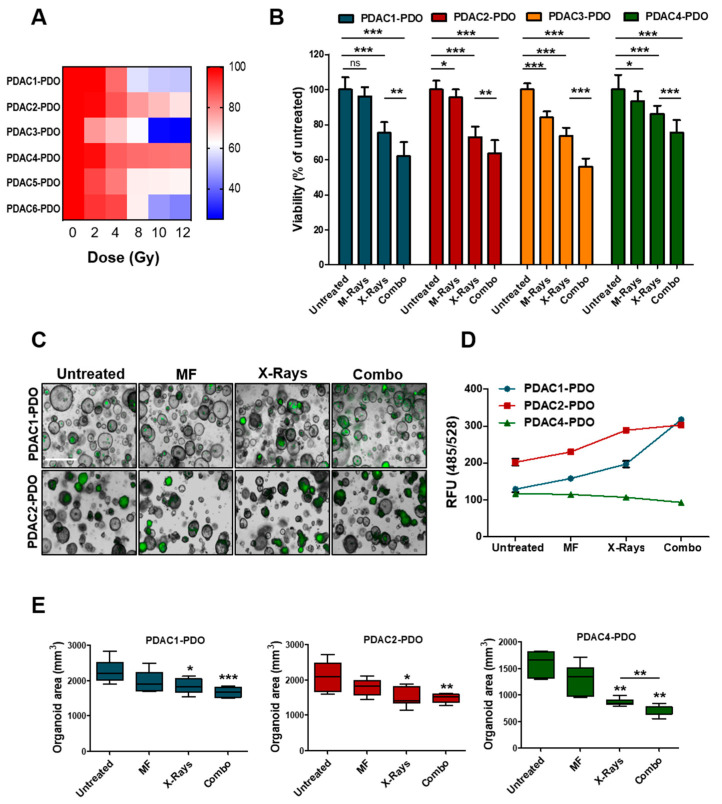
Response of PDAC-PDOs to Irradiation. (**A**) Heat-map showing dose-response to irradiation of six different PDAC-PDO cultures. Viability of each organoid culture as mean of five technical replicates. (**B**) Bar plots showing changes in cell viability after 6 Gy X-Rays in four representative PDAC-PDOs. Viability is reported as means of three independent experiments. (**C**) Representative images of organoids after 6 Gy X-Rays and MF, alone and/or in combination, compared to control. Apoptotic organoids are visualized as green structures. Scale bar, 100µm. (**D**) Quantification of green fluorescent signal (relative fluorescence units, RFU) in the 3 PDAC-PDO lines from C; data are displayed as mean ± SD. (**E**) Changes in oganoids size after MF, and 6 Gy X-Rays treatment, alone and/or in combination, compared to untreated organoids in the three organoid lines. Organoids size data shown are means from two independent experiments. *, *p* > 0.05; **, *p* < 0.01; ***, *p* < 0.001 by student’ *t* test.

**Figure 4 biomedicines-08-00609-f004:**
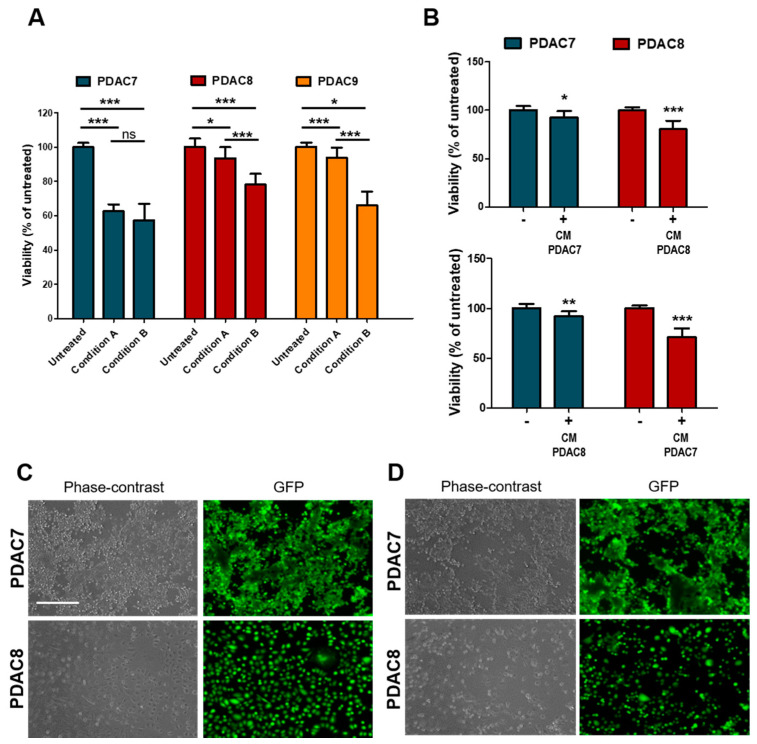
Response of 2D primary PDAC cultures to Irradiation. (**A**) Bar plots showing changes in cell viability after 6 Gy X-Rays in three primary 2D PDAC cultures under different conditions. Cell culture viability displayed as mean from three independent experiments. Condition A, cell culture medium was aspirated after treatment and medium replaced, no replaced of the treated culture medium in condition B. (**B**) Changes in cell viability of two primary PDAC cultures exposed to media from irradiated and non-irradiated cell lines. Cell culture viability are means from three independent experiments. (**C**,**D**) Representative Phase Contrast and Fluorescence microscopy images of PDAC cell lines exposed to conditioned media from untreated C and treated D cell lines as in B. Fluorescent signal for calcein was acquired after 30 min of incubation with CellTrace Calcein Green AM, using EVOS Cell Imaging System. Scale bar, 400 µm. *, *p* > 0.05; **, *p* < 0.01; ***, *p* < 0.001 by student *t* test.

**Table 1 biomedicines-08-00609-t001:** List of PDAC cultures used in this study.

Samples	Type of Culture
PDAC-1	Organoid culture
PDAC-2	Organoid culture
PDAC-3	Organoid culture
PDAC-4	Organoid culture
PDAC-5	Organoid culture
PDAC-6	Organoid culture
PDAC-8	2D primary culture
PDAC-9	2D primary culture
PDAC-10	2D primary culture

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
