# Peer review of "Combinatorial Effect of Magnetic Field and Radiotherapy in PDAC Organoids: A Pilot Study"

_biomedicines, 2020, doi:10.3390/biomedicines8120609_

Round 1

Reviewer 1 Report

The authors investigated the additional effect of a magnetic field (MF) to radiation and found an increase in apoptotic rate of organoids exposed to radiation. The authors claim that this is a pro-apoptotic effect of the static 1.5T MF and understanding of the biological mechanisms underlying this effect is ongoing.

                However, this is definitely not a novel mechanism, but RF-induced heating due to long-term exposure to MRI; a well-known phenomenon of MRI (e.g. Nadobny et al. Evaluation of MR-Induced Hot Spots for Different Temporal SAR Modes Using a Time-Dependent Finite Difference Method With Explicit Temperature Gradient Treatment; IEEE Trans Biomed Eng. 2007 Oct;54(10):1837-50). Hyperthermia, i.e. mild heating of tumours, is known to enhance the effectiveness of radiation, for example by inhibition of DNA-damage repair (e.g. Krawczyk et al. Mild hyperthermia inhibits homologous recombination, induces BRCA2 degradation, and sensitizes cancer cells to poly (ADP-ribose) polymerase-1 inhibition; Proc Natl Acad Sci U S A. 2011 Jun 14;108(24):9851-6). Hyperthermia has been applied clinically to a variety of tumour sites, including pancreatic cancer (Van der Horst et al. The clinical benefit of hyperthermia in pancreatic cancer: a systematic review; Int. J. Hyperthermia 2018 Nov;34(7):969-979), and randomized clinical trials are ongoing (e.g. Datta et al, "HEATPAC" - a phase II randomized study of concurrent thermochemoradiotherapy versus chemoradiotherapy alone in locally advanced pancreatic cancer; Radiat Oncol. 2017 Nov 21;12(1):183).

                Thus, the authors did not discover a new unknown mechanism, nor will this heating by the MR-linac be significantly beneficial in clinical treatments. The phantom used easily raises in temperature, but in patients blood flow will cause heat removal and dedicated hyperthermia systems are required to apply meaningful focused heating to enhance the radiation effect (Kok et al Heating technology for malignant tumors: a review; Int J Hyperthermia. 2020;37(1):711-741). Therefore, I recommend to reject this article.

Reviewer 2 Report

The manuscript by Nicosia et al. is investigating the interesting and novel topic of combinatorial effect of magnetic fields, as the used in MRI, and radiotherapy on pancreatic ductal adenocarcinoma organoid cultures. The thesis of the authors is elaborated well and the idea of the combined treatment is novel and potentially good strategy to sensitize PDAC cells to radiotherapy. With all the positive sides, the study has some limitations: the used patient sample size is very small (n=6) and in the separate experiments there have been discussed 3-4 samples (not clear what happens with the rest 2-3). If the study size is so limited, ‘pilot study’ should be included in the title.

Major comments. In addition to the small study size, other aspects of the study are staying unclear. Is the used in the clinic 1.5T MR-linac dangerous for the patient’s normal tissues as well? How they will overcome that problem if the normal tissue is also damaged? Why the authors have not included normal pancreatic cell line as a control? There are some inconsistencies in the abstract and the materials and methods – first in the abstract there are mentioned 6 samples, in materials and methods they are 9 and then in separate experiments only 3-4. Please correct or explain that. Is 6 Gy clinically relevant dose? And if so, is that dose delivered in one time or in fractions in the clinic?

If the MR exposure lasts 60 min, how the normal cell culturing conditions are kept (37C, 5%CO2) during that time? Do the authors have control experiments on that (without MR)?

Minor and figure-specific comments:

Throughout the text: the cell populations (PDA1-PDO, PDA2-PDO etc.) are not referred and explained well (the labels are changing from PDA to PDAC and vice versa and it is bit confusing). On p9 row 278 there appears PDAC10. What is 10 here if there have been included 9 patient samples? Are PDAC8 and 9 relevant to 8 and 9 patient, or are they same population as the used in some of the earlier PDA-PDO? I suggest the authors to include a table in materials and methods where could clarify the numbering of the samples and for what experiments they have been used.

row 36 demonstrated limited sensitivity.. What do you mean with limited sensitivity? Any quantification? Also in the next row 37 demonstrated activity? What activity exactly?

The conclusion is vague.

Round 2

Reviewer 1 Report

The authors argue that their hypothesis that magnetic field might exert a biological effect other than the known physical effect (i.e., heating) is based on numerous and solid evidences. This might be true, but the present experimental set-up does not support such a conclusion. If the authors want to demonstrate that there is an MF effect that can increase therapeutic efficacy, they should exclude the heating effect. A heating effect will be present in an experimental set-up, but will be very minor in a clinical setting, where perfusion will be an important heat removal mechanism. Thus, the small heating effect will not increase therapeutic efficacy and as mentioned before dedicated heating systems are required to realize a meaningful focused heating. When the main effect we are facing here in this paper, is due to very mild heating, it will not be relevant for the clinical treatment for this reason. However, if the main effect is indeed due to the MF, as claimed by the authors it could indeed be interesting, but to demonstrate that properly the heating effect should be excluded, which means that an extra experimental group is essential. This should be done by measuring the temperature during the MR+RT experiments and perform additional experiments with RT + heating, thereby ensuring the same thermal dose as achieved during MR exposure. Comparing these RT+heat results with MR+RT results will then show whether there is indeed a relevant MF effect that can increase therapeutic efficacy and should be further investigated. From the present experimental set-up no valid conclusions on the MF effect can be drawn, which means that the present manuscript learns us in fact nothing.

Author Response

            We would like to thank the reviewer for his/her informative critique. We are confident that we have addressed the issue raised and feel that this has further improved the quality of the manuscript.

            A copy of the manuscript where changes are indicated in blue coloured font (added text) and as grey-crossed out (eliminated text) has been uploaded as Additional Files for Review but NOT for Publication.

Please find below the point-by-point response.

Reviewer #1 Comment (major)

The authors argue that their hypothesis that magnetic field might exert a biological effect other than the known physical effect (i.e., heating) is based on numerous and solid evidences. This might be true, but the present experimental set-up does not support such a conclusion. If the authors want to demonstrate that there is an MF effect that can increase therapeutic efficacy, they should exclude the heating effect. A heating effect will be present in an experimental set-up, but will be very minor in a clinical setting, where perfusion will be an important heat removal mechanism. Thus, the small heating effect will not increase therapeutic efficacy and as mentioned before dedicated heating systems are required to realize a meaningful focused heating. When the main effect we are facing here in this paper, is due to very mild heating, it will not be relevant for the clinical treatment for this reason. However, if the main effect is indeed due to the MF, as claimed by the authors it could indeed be interesting, but to demonstrate that properly the heating effect should be excluded, which means that an extra experimental group is essential. This should be done by measuring the temperature during the MR+RT experiments and perform additional experiments with RT + heating, thereby ensuring the same thermal dose as achieved during MR exposure. Comparing these RT+heat results with MR+RT results will then show whether there is indeed a relevant MF effect that can increase therapeutic efficacy and should be further investigated. From the present experimental set-up no valid conclusions on the MF effect can be drawn, which means that the present manuscript learns us in fact nothing.

Response: We thank the reviewer for his/her valuable comment. We respectfully disagree with this reviewer about the relevance of heating as induced by MF; indeed, it is hard to find evidences for such an effect in the current literature, while there are numerous reports describing other biological effects of static magnetic field (see previous rebuttal). Nonetheless, we have performed additional experiments to assess heating in our experimental set-up and found that there is no increase of the temperature when culture plates are exposed to the treatment. We have measured the temperature of individual wells containing different volumes of solution and matrix using an infrared pyrometer. The median temperature at T0 (before starting the experiment) was 29.6 °C (range 28.7-31.5). The plates were placed within the MR gantry of the MR-linac for 60 minutes and then temperature was measured (T1), reporting a median value of 28.8 °C (range 28.3-30.7). Those values are far below the mild heating effect suggested by this reviewer to be therapeutically relevant for hyperthermia and achieved in the referred publications through direct heating. Therefore, we can conclude that there is no heating effect in our experimental set up. Furthermore, we would like to point out that the MR-linac is equipped with a ventilator to allow the machine working under controlled temperature, and patients during treatment are often covered with a warm blanket to avoid unpleasant body cooling. We have now included information about the measurement of the temperature during experiments in the Material and Method section of the revised version of the manuscript. Please, see page 4, lines 173-174.

Reviewer 2 Report

The revised version of the manuscript by Nicosi et al. has significant improvements. The authors had answered to all the questions and raised concerns thoroughly and the current version is in satisfactory shape to be published.

Author Response

Comments and Suggestions for Authors
-Reviewer #2

The revised version of the manuscript by Nicosi et al. has significant improvements. The authors had answered to all the questions and raised concerns thoroughly and the current version is in satisfactory shape to be published.

Response: We thank the reviewer for his/her very supportive comment. We are similarly excited by the possibility of publishing this work.